# Re-Emergence of DENV-3 in French Guiana: Retrospective Analysis of Cases That Circulated in the French Territories of the Americas from the 2000s to the 2023–2024 Outbreak

**DOI:** 10.3390/v16081298

**Published:** 2024-08-14

**Authors:** Alisé Lagrave, Antoine Enfissi, Sourakhata Tirera, Magalie Pierre Demar, Jean Jaonasoa, Jean-François Carod, Tsiriniaina Ramavoson, Tiphanie Succo, Luisiane Carvalho, Sophie Devos, Frédérique Dorleans, Lucie Leon, Alain Berlioz-Arthaud, Didier Musso, Anne Lavergne, Dominique Rousset

**Affiliations:** 1Arbovirus National Reference Center, Virology Unit, Institut Pasteur de la Guyane, Cayenne 97300, French Guiana; alagrave@pasteur-cayenne.fr (A.L.); antoine.enfissi@pasteur.fr (A.E.); stirera@pasteur-cayenne.fr (S.T.); anne.lavergne@pasteur.fr (A.L.); 2Laboratoire Centre Hospitalier de Cayenne, Cayenne 97300, French Guiana; magalie.demar@ch-cayenne.fr (M.P.D.); jean.jaonasoa@ch-cayenne.fr (J.J.); 3Department of Biology, West French Guiana Hospital Center, Saint-Laurent-du-Maroni 97320, French Guiana; jf.carod@ch-ouestguyane.fr (J.-F.C.); t.ramavoson@ch-ouestguyane.fr (T.R.); 4Santé Publique France, Cellule Guyane, Cayenne 97300, French Guiana; tiphanie.succo@santepubliquefrance.fr (T.S.); luisiane.carvalho@santepubliquefrance.fr (L.C.); sophie.devos@santepubliquefrance.fr (S.D.); 5Santé Publique France, Cellule Antilles, French Caribbean Islands; frederique.dorleans@santepubliquefrance.fr (F.D.); lucie.leon@santepubliquefrance.fr (L.L.); 6Laboratoires Eurofins Guyane, French Guiana; alain.berlioz@biologie.eurofinseu.com (A.B.-A.); didier.musso@biologie.eurofinseu.com (D.M.)

**Keywords:** DENV-3, re-emergence, DENV monitoring, NGS, molecular epidemiology

## Abstract

French Guiana experienced an unprecedented dengue epidemic during 2023–2024. Prior to the 2023–2024 outbreak in French Guiana, DENV-3 had not circulated in an epidemic manner since 2005. We therefore studied retrospectively the strains circulating in the French Territories of the Americas (FTA)—French Guiana, Guadeloupe, and Martinique—from the 2000s to the current epidemic. To this end, DENV-3 samples from the collection of the National Reference Center for Arboviruses in French Guiana (NRCA-FG) were selected and sequenced using next-generation sequencing (NGS) based on Oxford Nanopore Technologies, ONT. Phylogenetic analysis showed that (i) the 97 FTA sequences obtained all belonged to genotype III (GIII); (ii) between the 2000s and 2013, the regional circulation of the GIII American-I lineage was the source of the FTA cases through local extinctions and re-introductions; (iii) multiple introductions of lineages of Asian origin appear to be the source of the 2019–2021 epidemic in Martinique and the 2023–2024 epidemic in French Guiana. Genomic surveillance is a key factor in identifying circulating DENV genotypes, monitoring strain evolution, and identifying import events.

## 1. Introduction

Dengue, which is endemic in tropical and subtropical regions, is the most common and fastest-spreading mosquito-borne disease in the world. Over the past two decades, the World Health Organization (WHO) has reported a dramatic increase in the global incidence of dengue, from 505,430 cases in 2000 to 5.2 million in 2019 [1]. Although Asia usually accounts for most of the global disease burden, the Americas region is also experiencing an exponential rise in dengue cases. In 2023, with almost 4.6 million cases reported, the Americas accounted for 80% of the global rate: 7665 severe cases and 2363 deaths (case fatality rate 0.052%) were recorded in the region [1,2,3]. For at least two decades, the epidemiology of dengue in French Guiana and the other French Territories of the Americas (FTA)—Guadeloupe and Martinique—has been characterized by an endemo-epidemic circulation of dengue, with a 3–5-year periodicity of outbreaks, often associated with a shift in the predominant circulating dengue virus (DENV) serotype [4,5,6,7]. Since July 2023, French Guiana has been facing an unprecedented dengue epidemic, which was still ongoing in April 2024. According to Santé publique France (SpF), epidemic transmission is defined by a significant increase in the main surveillance indicators (number of confirmed cases and number of clinically evocative cases of dengue) for at least two consecutive weeks, reaching or exceeding the levels observed at the start of past epidemics. A total of 17,485 clinically evocative cases of dengue have been seen in general medical services or health centers, and 10,303 confirmed cases have been reported, with co-circulation of the two serotypes DENV-2 and DENV-3 [8]. Interestingly, prior to the 2023–2024 outbreak in French Guiana, DENV-3 had not circulated in an epidemic manner since 2005. DENV-3 is subdivided into five genotypes—genotypes I to V (GI–GV) [9,10]. Genotypes I and II are detected in Asian regions; genotype III is widely distributed (Asia, the Caribbean, the Americas, and Europe); genotype IV consists of old strains, no longer detected, isolated from Tahiti (1965) and Puerto Rico (1963/1977); and finally, genotype V comprises Asian strains and a few American strains, such as those from Brazil and Colombia [10].

Few DENV sequences from French Guiana are available in public databases, and no DENV-3 sequences from the 2023–2024 epidemic in French Guiana have been reported to date. However, studying the genetic evolution of DENV is essential for our understanding of the evolutionary history of strains circulating in the region, tracing importation events, and dating common ancestors. To address this lack of knowledge, we studied DENV-3 strains that circulated in the FTA, i.e., French Guiana, Guadeloupe, and Martinique, from the 2000s to 2024. A total of 97 complete DENV3 genomes were obtained and used in phylogenetic analyses to determine the circulating genotypes and to achieve a better understanding of the dynamics of DENV-3 circulation in FTA.

## 2. Materials and Methods

### 2.1. Ethics Statement

This study was part of a public health surveillance program of the National Reference Center for Arboviruses in French Guiana (NRCA-FG), which works in collaboration with the French Public Health Agency (Santé Publique France, SPF). Accordingly, as an epidemiological record, approval by an ethics committee was not required. The samples involved in this study were chosen among remaining human serum samples received as part of standard diagnostic and expertise activities of the NRCA-FG and stored in the NRCA-FG biobank according to French legislation (article L.1211–2 and related articles of the French Public Health Code—FPHC). Human serum samples were anonymized, with no or minimal risk to patient data privacy, in accordance with the terms of the European General Data Protection Regulation and the French National Commission on Informatics and Liberty (CNIL).

### 2.2. Clinical Samples and Study Design

The samples used in this study came from the NRCA-FG biobank, which comprises clinical specimens received from hospitals and private laboratories in French Guiana and the French Caribbean Islands as part of the arbovirus surveillance system. Between 2001 and January 2024, 2779 samples were tested DENV-3-positive, among which 1448 had a cycle threshold (Ct) value below 26 (10 from Guadeloupe, 75 from Martinique, and 1363 from French Guiana) (Table 1). Among these samples, a selection was made in each municipality and for each detection period, leading to 97 samples collected between 2001 and 2024 (Table 1) being selected for the analysis, including 5 from Guadeloupe, 7 from Martinique, and 85 from French Guiana. The location of these different municipalities is presented in Figure A1 of Appendix B.

### 2.3. MinION Library Preparation and Multiplexed Nanopore Sequencing

Whole-genome sequencing was performed using the MinION device (Oxford Nanopore Technologies, Oxford, UK) based on a protocol from Quick et al., 2017 [11]. Briefly, 8 μL of RNA was reverse transcribed to cDNA using LunaScript RT SuperMix (NEB, Ipswich, MA, USA) following the manufacturer’s instructions. A serotype-specific multiplex polymerase chain reaction (PCR) [12] was performed with Q5 High-Fidelity DNA polymerase (NEB), and the amplifications were performed into three distinct pools (Table A1). Primer schemes were adapted from the CDC protocol (Table A1) [13,14,15]. For the multiplex PCR, the cycling conditions were 30 s at 98 °C, followed by 35 cycles at 98 °C for 15 s, and 63 °C (Pool 1 and 2) or 60 °C (Pool 3) for 5 min.

The resulting PCR products were pooled, cleaned using AmpureXP magnetic beads (Beckman Coulter, Brea, California, USA), and quantified using a Qubit dsDNA High Sensitivity assay on a Qubit 3.0 instrument (Thermo Fisher Scientific, Waltham, MA, USA). The samples were then end-repaired using the Ultra II End Repair/dA-Tailing Module (New England Biolabs, Ipswich, MA, USA) and barcoded with the native barcoding kits NBD104 and NBD196 (Oxford Nanopore Technologies, Oxford, UK) according to the manufacturers’ instructions. They were subsequently cleaned with magnetic beads and pooled before ligation of the AMII adapters with blunt/TA ligase master mix (New England Biolabs) using the SQK-LSK-109 kit (Oxford Nanopore Technologies). Sequencing libraries were loaded onto the R9.4 flow cell (Oxford Nanopore Technologies), and sequencing data were collected overnight. Sequence reads were base-called and demultiplexed using the Guppy algorithm v3.6 (Oxford Nanopore Technologies). The consensus genome sequences were produced using the Artic network’s bioinformatics pipeline, version 1.2.4 (https://artic.network/ncov-2019/ncov2019-bioinformatics-sop.html, accessed on 1 February 2024), with a reference genome (MH544651.1) and the Medaka algorithm (https://github.com/nanoporetech/medaka, accessed on 1 February 2024). Regions with insufficient coverage (minimum depth coverage set to 20) were masked with N characters.

### 2.4. Phylogenetic Analysis

Whole-genome consensus sequences of DENV-3 were aligned with the CLC Main Workbench software (version 22; Qiagen, Hilden, Germany) to DENV-3 whole-genome reference sequences downloaded from GenBank. In total, 79 reference sequences were selected with a geographical and time distribution, covering the various genotypes described to date (Table A2). The most closely related sequences of each NRCA-FG-generated sequence were retrieved from GenBank based on a BLAST search (NCBI). The 97 sequences produced in this study and described in Table A3 were aligned with the 79 reference genomes of DENV-3 from the GenBank database. The BEAST and BEAUTI (version 1.10.1) software programs were used to construct the maximum clade credibility (MCC) tree, using the GTR + G + I (General Time Reversible with Gamma distribution and Invariant sites) substitution model, corresponding to the best-fit model tested on the CLC Main Workbench software, according to the corrected Akaike information criterion (AICc), with a strict clock model and Bayesian skyline prior [16,17,18]. A Markov chain Monte Carlo (MCMC) analysis was run for 50 million generations with sampling every 1000 generations. The final MCMC sampling chains were checked using Tracer v1.7.1, with 10% burn-ins removed. Convergence was obtained by reaching an effective sample size (ESS) of >200 for all parameters using Tracer v1.7.1 [19]. The MCC tree was generated with Tree Annotator 1.10.4, and the time-scaled phylogeny was visualized using FigTree v1.4.3 (http://tree.bio.ed.ac.uk/software/figtree/, accessed on 21 June 2024).

## 3. Results

The phylogenetic analysis was carried out on a dataset of 176 whole-genome DENV-3 sequences (Figure 1). All of the DENV-3 complete genome sequences obtained from French Guiana (FG), Guadeloupe (GLP), and Martinique (MTQ) having circulated between 2001 and 2024 belong to genotype III (GIII) (Figure 1).

### 3.1. Circulation of Regional Strains in the Early 2000s

Looking more specifically at GIII, the FTA sequences in the early 2000s (e.g., PP582621 FG 2001-11-04 and PP582622 FG 2013-01-24) are intermingled with other American sequences during the same period, including sequences from Venezuela in 2003, Mexico in 2007, Peru in 2008, and Brazil in 2002 and in 2007 (Figure 1). 

### 3.2. Multiple Introductions Are the Source of the 2019–2021 Epidemic in Martinique

DENV-3 sequences circulating in Martinique between 2019 and 2021 are divided into two clades: the first containing sequences from 2019 and the first half of 2020, and the second containing sequences from the second half of 2020. Sporadic cases of DENV-3 detected in Guadeloupe and French Guiana all belong to clade 2 (Figure 2). Clade 1 sequences shared 99.93–99.99% nucleotide identity/98.23–100% amino acid identity and are related to sequences from Africa (Ethiopia 2019 ON890788, MN964273). Clade 2 sequences shared 99.54–99.95% nucleotide identity/99.00–99.91% amino acid identity and are related to Asian sequence strains (Maldives 2019 ON890789, India 2016 MN018385). The percentage of nucleotide identity between these two clades ranges from 97.04% to 97.2%. A more detailed analysis of the nucleotide polymorphisms between these two clades was carried out (Table A4). The mutation rate varies throughout the coding genome, with the highest rates observed in the regions coding for the NS2A (3.21%) and NS5 (3.07%) proteins, Table A4. In addition, sequences from clade 1 shared 96.14–96.16% nucleotide identity with AY099337 Martinique 1999. The most recent common ancestor (MRCA) between AY099337 Martinique 1999 and clade 1 dates back to 1976 (95% HPD 1974–1979) (Figure 1), also suggesting a change of clade between them associated with a novel introduction.

### 3.3. New Introduction Is the Source of the 2023–2024 Epidemic in French Guiana

DENV-3 sequences from the 2023–2024 French Guiana epidemic are grouped in a well-supported monophyletic clade (bootstrap support 100%) with 99.52–100% nucleotide identity, and 99.65–100% amino acid identity. They are intermingled with sequences from the Americas, notably Brazilian sequences from the same year (OQ706226; OQ706227; OQ706228), Cuban sequences (OQ821545, OQ821510, OQ821537, OQ132878), or a Surinamese sequence (OQ868517). This clade, with an MRCA dating from 2020 (95% HPD 2019–2020), is clearly distinct from the one composed of the DENV-3 sequence circulating during 2020–2021. Indeed, the MRCA of these two clades dates back to 2007 (95% HPD 2006–2009). However, these 2023–2024 sequences are related to Asian strains (ON123669 India 2018, OM865820 Bhutan 2019, and India 2021 OM638675/ON109599).

### 3.4. Amino Acid Substitution Analysis among the 97 FTA Sequences

Amino acid (AA) substitution analysis between the 97 FTA DENV-3 sequences was carried out to assess the degree of amino acid variability over time. A total of 159 distinct amino acid substitutions was observed between sequences (Table A5). The regions with the highest number of substitutions are those coding for the envelope protein (22 AA changes), the NS2A protein (19 AA changes), NS3 (25 AA changes), and NS5, the RNA-dependent RNA polymerase (49 AA changes). The highest AA substitution rates are observed in regions coding for NS2A (8.72%), 2K (8.70%), NS2B (6.15%), and NS5 (5.51%) proteins. Of these AA changes, 51 are conserved across all strains in a given period. Some are period-specific, as observed for the 2023–2024 epidemic period, with six specific amino acid changes: 2530 I/V, 3487 F/L, 3931 M/L, 4946–4947 T/M, 5595 D/E, and 10,060 P/S. Others are shared between different periods: between the 2019–2020 clade 2 viruses and the 2023–2024 viruses for the three AA changes 1312 V/I, 6706 L/M, and 9958 V/I; or between all strains of the 2019–2021 Martinique and 2023–2024 French Guiana epidemics for the AA changes 2596 I/T, 2851 L/M, 4204 I/V, 4510 F/L, 4597 Y/H, 7069 I/V, 8032 T/A, 8155 A/S, 8563 S/P, 8591 G/E, and 9154 Q/L. It is therefore possible to define period-specific substitution profiles (2001/2013/2019–2021/2023–2024). The clades (clades 1 and 2) of the 2019–2021 Martinique epidemic thus can be distinguished from each other (clades 1 and 2) and from other FTA strains by a combination of 18 specific AA changes (Appendix A).

## 4. Discussion

In this study, we described the DENV-3 genotypes circulating in the FTA during a 20-year period as well as the recent introduction event leading to the current epidemic in French Guiana. The 97 FTA sequences obtained in this study all belong to DENV-3 GIII, which is therefore the only sequence to have been detected in French Guiana, Guadeloupe, and Martinique between the 2000s and the present. In addition, analyses of amino acid substitution profiles by period for the 97 sequences in the FTA (2001/2013/2019–2021/2023–2024) clearly show the genetic evolution over time of the DENV-3 virus circulating in this region. According to the literature, DENV-3 GIII circulates worldwide, while the other genotypes are localized in particular geographical areas [15,20,21]. This genotype has been implicated in major dengue epidemics in several regions of Asia, America, and Africa [15,20,21,22,23,24,25,26,27], which suggests that it has great potential to spread and adapt in various geographical regions of the world [15,20,22,23]. First introduced in Central America in the 1990s from Asia, the DENV-3 GIII sequence rapidly spread and expanded into the Americas [15]. The extensive transmission across the Americas observed in the subsequent years, combined with a significant separate evolution of this genotype from the Asian lineage, led to a new lineage: the DENV-3 GIII American-I lineage [15]. In the FTA, DENV-3 were first described in 1999, circulating at high levels until 2005 before falling back to low levels, with only a few sporadic cases reported over the next 15 years [5,28,29]. This study shows that between 1999 and 2013, all the FTA strains belong to the DENV-3 GIII American-I lineage, but to distinct clades, intermingled with other American sequences circulating during the same period. These observations favor the notion of a circulation in the FTA based on extinctions and re-introductions of strains from regional circulating strains rather than the re-emergence of previously circulating strains. Even though the DENV-3 GIII American-I lineage was still recorded in Mexico in 2021 [15], the diversity of DENV-3 GIII circulating in the Americas has increased significantly in recent years by introductions of new GIII lineages, as shown in this study. These multiple introductions are associated with a large re-emergence of DENV-3 in the region and have led to recent epidemics, such as the 2019–2021 outbreak in Martinique or the 2023–2024 outbreak in French Guiana.

Surprisingly, the DENV-3 sequences from the 2019–2021 Martinique epidemic belonged to two different clades, highlighting multiple introductions of DENV within a single epidemic. These two clades—the first grouped with sequences originating from Africa and the second with sequences from Asia—have already been described by Garcia Van Smévoorde et al. [30]. Interestingly, the two clades succeeded one another during the epidemic, with the first clade detected from the end of 2019 to mid-2020, associated with a moderate epidemic, which was replaced after mid-2020 by the emergence of clade 2, associated with an unprecedented epidemic peak. Overall, this was the longest epidemic (67 weeks) ever recorded in Martinique since surveillance began, the most intense, and also the most severe [31].The highest rates of signature mutations between clades 1 and 2 are observed in the regions coding for the NS2A (3.21%) and NS5 (3.07%) proteins. As the NS5 protein plays a key role in viral replication [32], this leads us to question the involvement of these mutations in the intensity and severity of the second epidemic phase. The intensity and the length of the epidemic may have been favored by multiple introductions and successive circulations.

The DENV-3 sequences from the 2023–2024 French Guiana epidemic appeared clearly distinct from both clades identified in the FTA during 2019–2020. These FG sequences belong to a well-supported monophyletic clade and are intermingled with sequences from the Americas, notably from Brazil, Cuba, and Suriname with which they share a recent common ancestor (2020). All of this suggests a recent introduction and circulation of this lineage in the American region. The closest sequences of non-American strains identified in NCBI come from Asia (India, Bhutan), suggesting an Asian origin through one or more introductions. However, other routes of introduction cannot be excluded, as these data depend on what is available in databases, which are not exhaustive.

These observations highlight the intensity of the exchanges of DENV-3 strains at a regional but also global level. In a similar way, Klitting et al. have described recent and multiple introductions of DENV-2 in the FTA [33]. The intensity of these exchanges has an impact on the diversity of DENV and on the epidemiological evolution.

The DENV-3 extinction observed between 2005 and 2023–2024 in FG could be partly explained by the population’s herd immunity caused by the dominant circulation of the DENV-3 GIII American-I lineage during the 2001–2002 and 2004–2005 epidemics. The epidemics that followed were then associated with the other serotypes, with a classic alternation of the predominant serotypes: the 2006 epidemic was associated with a majority of DENV2, the 2009–2010 epidemic with a majority of DENV-1 and a co-circulation of DENV-4, the 2012–2013 epidemic with a majority of DENV-2 and a co-circulation of DENV-4, and the 2019–2020 epidemic with a majority of DENV-1 and a co-circulation of DENV-2 strains. The long absence of DENV-3 circulation may also have been favored by the absence of a significant circulation of DENV from 2014 to 2019. This absence, unusual in endemo-epidemic regions for DENV, was concomitant with the emergence of CHIKV and ZIKV, suggesting potential viral competition for the vector Aedes aegypti [34,35,36,37].

The re-emergence of DENV-3 in FG, combined with a co-circulation of DENV-2, led to the unprecedented epidemic of 2023–2024 [34,38]. An increase in the dengue-naive population resulting from, for example, tourism in Martinique and Guadeloupe or the high birth rate in French Guiana could have favored the re-emergence of DENV-3 in the FTA [39,40,41,42,43,44].

However, a new lineage is not necessarily associated with an epidemic, as shown during 2019–2020, with the importation of another new DENV-3 GIII lineage responsible for a major epidemic in Martinique but only for sporadic cases in FG. Thus, other factors are involved in the spread and diversification of DENV, such as environmental and socioeconomic factors facilitating the adaptation and proliferation of vectors, or alterations in the population’s herd immunity resulting from changes in the level of immunity [35,45,46,47,48,49].

Infection with any DENV serotype results in long-term homotypic immunity [50]. However, the increase in genetic diversity associated with the introduction of new lineages may lead us to wonder about the impact of the genetic diversity on antigenic characteristics and therefore on the immunity induced by previous infections and their potentially protective or facilitating nature. This point is key for defining vaccine development strategies [46,50,51,52,53,54,55,56].

## 5. Conclusions

In conclusion, our findings showed that the sequences obtained in this study all belong to DENV-3 GIII, with distinct lineages reflecting intense exchanges in the Americas and also worldwide (with import particularly from Asia). Indeed, over a period of 20 years and especially in the last 5 years, multiple introductions have been identified. This study highlights the importance of genomic surveillance and sequencing efforts in the FTA, but also globally, for monitoring the emergence and re-emergence of DENV-3 and, more generally, of all serotypes. Studying the genetic evolution of DENV is therefore essential for understanding the evolutionary history of strains circulating in the region and for tracing import events. These data are important public health elements for monitoring epidemics.

## Figures and Tables

**Figure 1 viruses-16-01298-f001:**
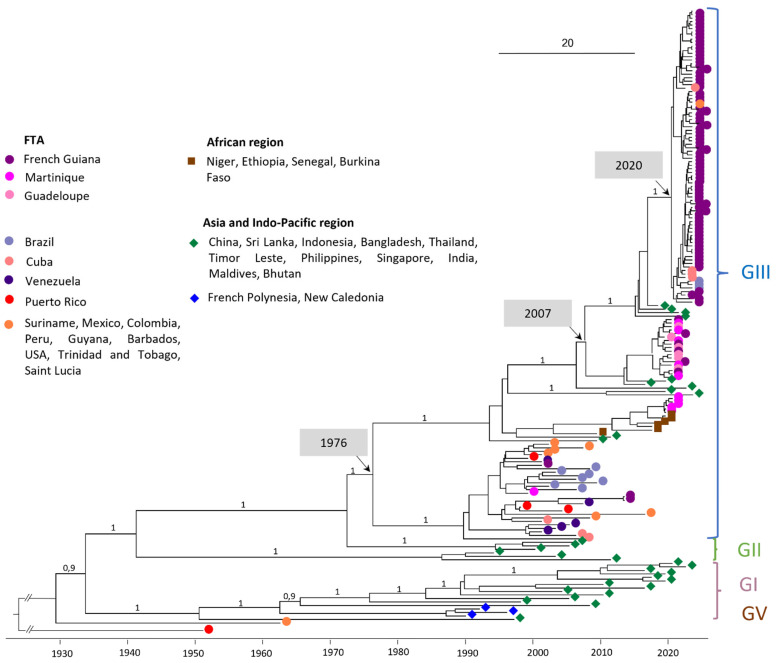
Bayesian phylogeny of DENV-3 whole-genome sequences. The maximum clade credibility (MCC) tree, using the GTR + G + I (General Time Reversible with Gamma distribution and Invariant sites) substitution model, was constructed with a strict clock model and Bayesian skyline prior. The analysis included 97 sequences obtained with nanopore sequencing technology (Oxford Nanopore Technologies, ONT) at NRCA-FG and 79 reference sequences downloaded from GenBank. DENV-3 OM258630 Puerto Rico 1953 (unclassified genotype) was used as the outgroup.

**Figure 2 viruses-16-01298-f002:**
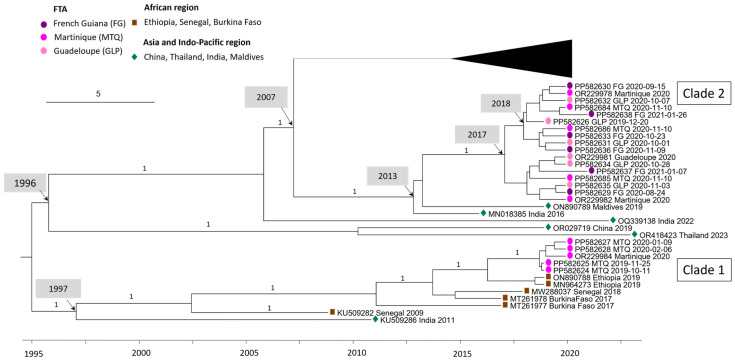
Bayesian phylogeny of DENV-3 whole-genome sequences with a focus on the 2020 Martinique outbreak. The maximum clade credibility (MCC) tree, using the GTR + G + I (General Time Reversible with Gamma distribution and Invariant sites) substitution model, was constructed with a strict clock model and Bayesian skyline prior.

**Table 1 viruses-16-01298-t001:** Number of generated sequences (N selected) by territory/municipality and year of sampling among samples found positive for DENV-3 with a Ct < 26.

Territory	Total N with Ct < 26	Municipality	Year (s)	N Selected	Municipality	Year(s)
**Guadeloupe**	**10**			**5**		
	3	Baie-Mahault	2019/2020	2	Baie-Mahault	2019/2020
	4	Le Gosier	2020	2	Le Gosier	2020
	1	Les Abymes	2020	1	Les Abymes	2020
	2	Other	2020			
**Martinique**	**75**			**7**		
	14	Le Vauclin	2019	1	Le Vauclin	2019
	5	Les Trois-Îlets	2019/2020	2	Les Trois-Îlets	2019/2020
	1	Les Anses-d’Arlet	2019	1	Les Anses-d’Arlet	2019
	4	Le Robert	2020	1	Le Robert	2020
	10	Fort-de-France	2019/2020	1	Fort-de-France	2020
	11	Schoelcher	2019/2020	1	Schoelcher	2020
	30	Other	2019/2020			
**French Guiana**	**1363**			**85**		
	119	Cayenne	2001/2013/2020/2021/2023/2024	11	Cayenne	2001/2013/2021/2023
	678	Kourou	2022/2023/2024	20	Kourou	2022/2023/2024
	251	St-Laurent-du-Maroni	2023/2024	12	St-Laurent-du-Maroni	2023/2024
	90	Remire-Montjoly	2020/2023/2024	7	Remire-Montjoly	2020/2023/2024
	37	Tonate-Macouria	2020/2023/2024	3	Tonate-Macouria	2020/2023/2024
	10	Sinnamary	2023/2024	4	Sinnamary	2023/2024
	3	Iracoubo	2023	1	Iracoubo	2023
	16	Mana	2023/2024	5	Mana	2023
	17	Maripasoula	2023/2024	6	Maripasoula	2023
	58	Matoury	2013/2020/2023/2024	7	Matoury	2020/2023/2024
	4	Saint-Georges	2023	1	Saint-Georges	2023
	2	Antecum Pata	2023	1	Antecum Pata	2023
	1	Montsinery	2023	1	Montsinery	2023
	26	Grand-Santi	2023/2024	3	Grand-Santi	2023
	10	Apatou	2023/2024	2	Apatou	2023
	3	Papaïchton	2023	1	Papaïchton	2023
	38	Other	2020/2023/2024			
**Total**	**1448**			**97**		

## Data Availability

All virus sequences are accessible on GenBank (accessions specified in the Appendix A).

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
