# Peer review of "Re-Emergence of DENV-3 in French Guiana: Retrospective Analysis of Cases That Circulated in the French Territories of the Americas from the 2000s to the 2023–2024 Outbreak"

_viruses, 2024, doi:10.3390/v16081298_

Round 1
Reviewer 1 Report
Comments and Suggestions for Authors
The authors identified an unprecedented dengue fever outbreak in French Guiana during 2023-2024, with the DENV-3 serotype not having circulated epidemically since 2005. Therefore, the authors aim to investigate the characteristics of these epidemic DENV-3 strains. Upon thorough review, several issues remain with this manuscript. First, the specific scientific question the authors seek to address by studying these strains is not clearly defined. Secondly, the description of the experimental design is imprecise, making it difficult to understand the authors' main focus. Lastly, the analysis is overly simplistic, relying solely on phylogenetic tree results, which do not adequately explain the transmission dynamics of the strains. In summary, the authors need to clearly define the manuscript's focus, provide additional results, and improve the logical flow of the article.
Abstract
Line 15: Specify what constitutes epidemic transmission.
Line 16: Explain the rationale for choosing the French Territories of the Americas (FTA).
Line 19: Provide details on the sample size and collection time points.
Materials and Methods
Line 79: Clarify what is meant by "randomly selected positive samples." Why were all positive samples not included?
Results
1. The authors should conduct a separate analysis of the 2023-2024 data to highlight the study's focus.
2. The authors could refer to the paper with PMID: 31671157 to create a transmission map.
Lines 156-160: Clarify the conclusions intended to be drawn from these findings.
Line 166: Explain why it is impossible for viruses from other regions to have been introduced to French Guiana. Systematic geographic analysis is needed to support this conclusion.
Reviewer 2 Report
Comments and Suggestions for Authors
The manuscript authored by Lagrave and colleagues presents findings about the re-emergence of DENV-3 in French Guiana. Thus, the authors studied retrospectively the DENV strains circulating in the French Territories of the Americas (FTA) – French Guiana, Guadeloupe, and Martinique – from the 2000s to the 2023.
Therefore, this manuscript by Lagrave et al. has the major findings:
“1) Phylogenetic analysis showed that the 97 FTA sequences obtained all belonged to genotype III (GIII);
2) Between the 2000s and 2013, the regional circulation of the GIII American-I lineage was the source of the FTA cases through local extinctions and re-introductions.
3) Multiple introductions of lineages of Asian origin appear to be the source of the 2019–2021 epidemic in Martinique and the 2023–2024 epidemic in French Guiana.”
The authors' study is relevant in the context of genomic surveillance of circulating DENV genotypes and provides important data on arboviral circulation in a region lacking this type of study (i.e., few DENV sequences are available in public databases). However, as highlighted below, overall, I provide suggestions that can improve the study.
1. Introduction (line 42): Please considerer writing “dengue virus (DENV) serotype” instead of “dengue virus serotype”
2. Material and Methods: Consider including a location figure for context.
3. I find it interesting that the authors perform amino acid sequence alignments alongside organizational comparisons. These analyses will enable us to assess the degree of conservation amino acid over time for DENV-3 GIII (i.e., over 20 years). Select viral protein(s) and protein regions that are most relevant in an infection context.
Reviewer 3 Report
Comments and Suggestions for Authors
In this article Alisé Lagrave and coll. did a retrospective analysis of the genotype of Dengue-3 cases observed within French West-Indies to French Guyana from 2000 up 2024. By using NGS sequencing of 97 samples randomly selected within the DENV-3 confirmed cases of the arboviruse surveillance program.
The description of the work is clear and there were no confusing part. The conclusion from the authors are clearly online with the obtained data. The analysis is nicely presented and convincing.
My first and quite to be the single comment I have was : we would like to know how many DENV-3 pos samples were within the biobank in the respective area from table I ? This table deserved to be completed with this number / total positive per municipality, number with Ct<26 and as shown number selected. This should be detailed a bit within the introduction I feel.
As a second comment, did the author try to define a sequence signature for the different clades and specifically the two clades described in the chapter 3.3. Perhaps this was done by Garcia Van Smévoorde et al. (line 207). To note is there really in link between Garcia and Van or is it a typo mistake ?
Round 2
Reviewer 1 Report
Comments and Suggestions for Authors
All concerns have been addressed